# An Ultra-Thin, Microwave-Absorbing Wear Layer for Pavement Deicing

**DOI:** 10.3390/ma16083080

**Published:** 2023-04-13

**Authors:** Xiaoming Liu, Fei Chang, Yu Zhao

**Affiliations:** School of Civil Engineering, Central South University, Changsha 410075, China

**Keywords:** asphalt concrete, ultra-thin microwave-absorbing wear layer, microwave deicing, energy saving

## Abstract

Microwave heating is widely employed in pavement deicing. However, it is difficult to improve the deicing efficiency because only a small part of the microwave energy is used and most of it is wasted. To improve the utilization efficiency of microwave energy and the deicing efficiency, we used silicon carbide (SiC)–replaced aggregates in asphalt mixtures to prepare an ultra-thin, microwave-absorbing wear layer (UML). The SiC particle size, SiC content, oil–stone ratio and thickness of the UML were determined. The effect of the UML on energy saving and material reduction was also evaluated. Results show that only a 10 mm UML was needed to melt a 2 mm ice layer within 52 s at −20 °C and rated power. In addition, the minimum layer thickness to meet the specification requirement (≥2000 με) of asphalt pavement was also 10 mm. SiC with larger particle sizes increased the temperature rise rate but decreased the temperature uniformity, instead increasing the deicing time. The deicing time of a UML with SiC particle size less than 2.36 mm was 35 s shorter than that of a UML with SiC particle size greater than 2.36 mm. Furthermore, more SiC content in the UML resulted in a higher temperature rise rate and less deicing time. The temperature rise rate and deicing time of the UML with 20% SiC were 4.4 times and 44% of those of the control group. When the target void ratio was 6%, the optimum oil–stone ratio of UML was 7.4%, and it had good road performance. Compared to overall heating, the UML saved 75% of power and SiC material under the same heating efficiency. Therefore, the UML reduces microwave deicing time and saves energy and material.

## 1. Introduction

Compared with traditional deicing methods, microwave deicing has attracted great interest from industry and academia thanks to its advantages of no damage to pavement and environmental friendliness [1,2]. However, the microwave heating performance of asphalt pavement is often improved by overall microwave enhancement, which limits the improvement of deicing efficiency [3]. Meanwhile, melting ice only requires heat from the surface of the pavement, the internal heat is invalid and results in energy and material waste [4]. Therefore, it is necessary to find a suitable pavement structure that can improve the deicing efficiency by accumulating heat on the surface.

The use of a thin-layer structure on asphalt pavement can improve the temperature rise rate of the road surface. Peng prepared a superhydrophobic coating with microwave heating function and proved that the temperature rise rate of the coated asphalt mixture was 48.37% faster than that of the uncoated asphalt mixture [5]. Additionally, the more heat accumulates on the surface, the more efficient the ice melting will be. Wan designed an ultra-thin friction course using steel slag and studied its ice and snow melting efficiency under induction heating. The results showed that the addition of steel slag reduced the effective induction heating depth and improved the snow and ice melting efficiency of the pavement [6]. Liu used epoxy resin as a binder to prepare a microwave-enhanced functional layer and conducted a microwave deicing test. The results indicated that the microwave-enhanced functional layer could effectively shorten the deicing time [3]. Moreover, accumulating heat on the surface also has the effect of saving energy and reducing emissions. Liu accumulated the heat on the pavement surface by adding waste steel wool into the asphalt mixture, which saved the energy required for microwave deicing and reduced the emissions of SO_2_ and NO_X_ [7]. An ultra-thin wear layer is an asphalt pavement thin overlay that has emerged in recent years and has a wide range of applications [8,9]. It can improve the skid and wear resistance of the pavement as well as repair the original pavement microcracks [10]. Therefore, the combination of ultra-thin wear layer and microwave deicing is beneficial for the further promotion and application of microwave deicing.

Scholars often improve the microwave deicing efficiency of pavement by adding microwave-absorbing materials [11,12]. According to the type of loss, the incorporated microwave-absorbing materials can be divided into magnetic loss materials, such as magnetite, and dielectric loss materials, such as activated carbon powder (ACP) [13,14]. The addition of 80% magnetite aggregate to the asphalt mixture increased the temperature rise rate to 0.41 °C/s and significantly reduced the deicing time [15]. Liu partially replaced the filler in asphalt mixture with ACP and tested the ice melting efficiency, showing that the addition of ACP increased the ice melting efficiency by 2.47 times [16]. Sun added steel slag as aggregates into the asphalt mixture to improve its ice melting performance and proved that the ice melting rate can be increased to 18.5 g/min by adding steel slag [17]. Besides magnetite and steel slag, other microwave-absorbing materials that have been studied include steel fiber, activated carbon powder, graphite and carbon black [18,19,20]. Although the addition of these microwave absorbers can improve the deicing efficiency of asphalt mixture, it also has the disadvantage of lowering the engineering performance of asphalt concrete, making construction challenging and high cost. SiC as a typical dielectric material can be added to asphalt mixture to effectively increase the temperature rise rate of the asphalt mixture under microwave heating [21]. In addition, SiC is frequently utilized to improve the wear resistance of concrete surfaces because of its wear resistance, high hardness and chemical stability [22]. Therefore, adding SiC into the ultra-thin wear layer can improve its microwave-absorbing performance and ensure its wear resistance.

In this work, a UML was designed to improve the deicing efficiency of asphalt pavement. The thickness range of the UML was calculated first, and then the thickness of the UML was recommended using the low-temperature splitting test. The effect of temperature uniformity and temperature rise rate on the deicing efficiency of the UML was investigated by microwave heating test and microwave deicing test. Based on the test results, the particle size and content of SiC in the UML were determined. Meanwhile, the key factors affecting deicing time were analyzed. The oil–stone ratio of the UML was designed and tested. Finally, the energy and material saving effects of the UML were evaluated.

## 2. Materials and Methods

### 2.1. Materials

The bitumen was a high-viscosity, modified bitumen produced in Hunan, China. Its properties are shown in Table 1.

The aggregate was limestone. Its properties are shown in Table 2; they meet the Chinese specification for the Test Methods of Aggregate for Highway Engineering (JTG E42-2005).

The SiC used in this study was produced in Henan, China. After sieving, three different particle sizes of SiC were obtained: 0~2.36 mm, 0~9.5 mm and 2.36~9.5 mm. The appearance of SiC is very similar to the aggregate used in asphalt mixture, which suggests that SiC may adhere to asphalt well and form an interlocking structure with the aggregate.

### 2.2. Low-Temperature Splitting Test

#### 2.2.1. Sample Preparation

As an asphalt pavement thin overlay, the ultra-thin wear layer is required to be water resistant, skid resistant, wear resistant and crack resistant and to have high bonding strength. However, the thinness of the ultra-thin wear layer causes it to crack easily in service [23]. Furthermore, the asphalt pavement layer has poorer resistance to cracking in low-temperature environments. To investigate the influence of the UML thickness on low-temperature crack resistance. The specimens were Marshall specimens with a double-layer structure to fit the situation of the UML paved on the asphalt pavement surface. The upper layer was prepared with asphalt mixture for the UML, and its grading curve is shown in Figure 1. AC-13 asphalt mixture was used to prepare the lower layer. The UML thickness determined the upper layer height of the specimens, and the total height of the specimens was 63.5 mm. The diagram of the specimen structure is shown in Figure 2.

The lower layer was formed according to the Chinese specification for Standard Test Methods of Bitumen and Bituminous Mixtures for Highway Engineering (JTG E20-2011). The upper layer was made on the surface of the lower layer after it had been horizontally placed for 12 h. After demolding, each group of specimens was obtained.

#### 2.2.2. Test Process

The low-temperature splitting test was carried out on specimens with different UML thicknesses and control group specimens according to the Chinese specification for Standard Test Methods of Bitumen and Bituminous Mixtures for Highway Engineering (JTG E20-2011). Control group specimens were standard Marshall specimens formed at one time. All prepared specimens were frozen in a −10 °C low-temperature chamber for 24 h and then loaded with a multifunctional pavement material strength tester at a 1.0 mm/min loading rate. According to the maximum test load at which the specimens were damaged, the low-temperature splitting tensile strength of specimens was calculated using Equation (1).
(1)$$R_{T}=\frac{0.006287P_{T}}{h}$$
where $R_{T}\;is\;the\;splitting\;tensile\;strength\;(MPa)$, $P_{T}\;is\;the\;maximum\;test\;load\;(N)$ and *h* is the height of the specimen (mm).

### 2.3. Low-Temperature Bending Test

To ensure that the low-temperature crack resistance of UML met the specification requirements, a low-temperature bending test was conducted on specimens with different UML thicknesses according to the Chinese specification for Standard Test Methods of Bitumen and Bituminous Mixtures for Highway Engineering (JTG E20-2011). The specimens were 250 mm × 30 mm × 35 mm beams. The specimens consisted of two asphalt mixtures, the lower layer was an AC-13 asphalt mixture and the upper layer was an asphalt mixture for UML. The height of the upper layer was determined by the UML thickness and the height of the lower layer was 35 mm minus the UML thickness. At −10 °C, a pressure testing machine was used to load the specimens at a 50 mm/min loading rate. After the loading, the mid-span deflection at the time of specimen damage was recorded. Equation (2) was used to calculate the failure strain of the specimen.
(2)$$\varepsilon_{B}=\frac{6hd}{L^{2}}$$
where $\varepsilon_{B}\;is\;the\;failure\;strain\;(\mu \varepsilon )$, *h* is the height of specimen at mid-span section (mm), *d* is the mid-span deflection of the specimen (mm) and *L* is the span of the specimen (200 mm).

### 2.4. Microwave Heating Test

A microwave heating test was carried out to investigate the effect of SiC particle size and content on the temperature uniformity and temperature rise rate of the UML under microwave heating. The UML was prepared by mixing SiC in different particle sizes and contents with bitumen, aggregate and mineral powder. Except for their height, the specimens for the microwave heating test were similar in structure and formation to those in the low-temperature splitting test. The upper layer of the specimens was a 10 mm UML and the lower layer was 31.8 mm AC-13 asphalt concrete. A microwave oven with 900 W output power and 2.45 GHz frequency was used for the test. The prepared specimens were put in the microwave oven at room temperature. The microwave heating time was 30 s, and the surface temperature was recorded every 10 s by an infrared thermal imager. The recorded temperature was entered into the FLIR tools for processing and analysis and provided a temperature matrix of the specimen surface. The process is shown in Figure 3.

### 2.5. Microwave Deicing Test

In order to investigate the effect of SiC particle size and content on the deicing efficiency of UML, a microwave deicing test was carried out with specimens in microwave heating test. Briefly, 10 mm ice layers were made on the surface of the specimens. The thickness of the ice layers was controlled by water volume. A sealing belt was used to completely wrap the sides of the specimens to prevent any water from escaping during the making of the ice layers. The water was frozen on the surface of the specimens five times to ensure sufficient bonding between the ice layers and the specimens. The specimens with ice layers were frozen in a low-temperature chamber at −5 °C, −10 °C and −15 °C for 24 h. The low-temperature chamber temperatures were considered as initial temperatures.

The frozen specimens were taken out of the low-temperature chamber and placed vertically in a microwave oven. As soon as the microwave oven was turned on, the ice layers gradually melted as the surface temperature of the specimens rose. Gravity caused the ice layers to fall off the specimens after they had partially melted. The microwave oven was turned off when the ice layer fell off the specimens. The time taken for the microwave oven to heat the specimens was the time required for the ice layer to fall off. The time taken for the ice to fall off, the mass of melted ice and the temperature of the specimen surface were recorded. The time taken for the ice to fall off was recorded as the deicing time. The process of the microwave deicing test is shown in Figure 4.

### 2.6. Oil–Stone Ratio Design Test

The oil–stone ratio significantly affects the high-temperature stability, water stability, skid resistance and other road performance of the UML, which is an important technical indicator for the UML. The Marshall test was used to determine the optimum oil–stone ratio of the UML according to the Technical Specifications for Construction of Highway Asphalt Pavements (JTG F40-2004) in China. Four groups of oil–stone ratios of 6.5%, 7.0%, 7.5% and 8.0% were first selected. After measuring the bulk density of each group of specimens, the porosity (VV), voids in mineral aggregates (VMA), and saturation (VFA) were calculated. Following the determination of optimum oil–stone ratio, the Schellenberg binder drainage test, Cantabro test, immersion Marshall test, freeze–thaw splitting test and high-temperature rutting test were used to check the design of the asphalt mixture.

## 3. Results and Discussion

### 3.1. Thickness of UML

#### 3.1.1. Determination of Thickness Range

The minimum thickness of the UML was calculated first. There are three stages to the microwave deicing process. Microwaves penetrate the ice layer and heat the asphalt pavement to 0 °C in the first stage, while the ice layer is still frozen [11]. In the second stage, the temperature of the asphalt pavement continues to rise, and the ice layer absorbs heat from the pavement and gradually melts. In the third stage, the ice adhering to the road surface melts into water causing the ice to separate from the surface. The ice needs to absorb a large amount of heat from the pavement in order to melt into water in the second stage [19]. Therefore, when the microwave heating time is certain, if the UML is too thin, it will not be able to store enough heat to melt the ice during the microwave heating. The minimum thickness of the UML was calculated using Equations (3)–(5).
(3)$$c_{a}m_{a}vt=am_{i}+c_{w}m_{i}T+c_{a}m_{a}T$$
(4)t=1.2T+28
(5)$$h_{min}=\frac{m_{a}}{\rho }\times 1000$$
where *c_a_* is the specific heat capacity of asphalt mixture (J/(kg·°C)), *m_a_* is the mass of asphalt mixture per unit area (kg/m^2^), *v* is the temperature rise rate of the UML (°C/s), *t* is the deicing time (s), *a* is the melting heat of ice (3.34 × 10^5^ J/kg), *m_i_* is the mass of ice melted per unit area (kg/m^2^), *c_w_* is the specific heat capacity of water (4200 J/(kg·°C)), *T* is the absolute value of ambient temperature (°C), *ρ* is the density of asphalt mixture (kg/m^3^) and *h_min_* is the minimum thickness of the UML (mm).

The values used in the calculation are shown in Table 3 [3,7,24,25]. According to the calculation, the minimum thickness of the UML required to be paved was 8.5 mm and 9.4 mm under ambient temperatures of 0 °C and −20 °C, respectively. The above data show that the additional heat required for deicing due to the lower ambient temperature is a very small fraction of the total heat required for deicing. Most of the heat is used in the phase-change process of ice melting from solid to liquid at the same temperature. Equations (3)–(5) indicate that, the lower the ambient temperature, the more the heat required for microwave deicing and the greater thickness of the UML needed to be paved. To meet the deicing requirements of the UML under different ambient temperatures, the minimum thickness of the UML required to be paved under −20 °C was considered as the minimum thickness required for UML deicing. Considering the paving process, the minimum thickness of the UML was set at 10 mm.

The maximum thickness of the UML was then calculated. The heating depth in microwave deicing is related to the output power of the microwave vehicle, the frequency of the microwave and the wave absorption performance of the asphalt concrete. When the output power and microwave frequency are known, the better the microwave absorption performance of asphalt concrete and the smaller the heating depth of microwaves [7]. Because UML has better microwave absorption properties than ordinary asphalt concrete, the microwave heating depth in the UML will be greatly reduced. To determine the maximum thickness of the UML, the maximum heating depth of a 2.45 GHz microwave in the UML was calculated using Equations (6) and (7).
(6)$$P=c_{a}m_{a}v$$
(7)$$h_{max}=\frac{m_{a}}{\rho }\times 1000$$
where *P* is the output power per unit area of microwave heating vehicle (kW/m^2^) and *h_max_* is the maximum thickness of the UML (mm).

In the calculation, the output power per unit area of the microwave heating vehicle was 60 kW/m^2^. According to the calculation result, the maximum thickness of the UML was 23 mm. In summary, the thickness of the UML ranged from 10 mm to 23 mm. Based on the layer thickness range, the UML used the gradation shown in Figure 2 with a maximum aggregate particle size of 9.5 mm.

#### 3.1.2. Crack Resistance of the UML

Based on the thickness range of the UML calculated above, 10 mm, 15 mm, and 20 mm UML specimens and control specimens were prepared. The low-temperature splitting test and low-temperature bending test were then performed on each group of specimens. The splitting tensile strength of each group of specimens is shown in Figure 5.

It can be seen from Figure 5 that the low-temperature splitting tensile strength of the UML specimens was consistently lower than that of the control specimens. This means that the UML would weaken the overall crack resistance of asphalt pavement. This is because, with the addition of the UML, the specimens resisting the splitting loads changed from an overall structure to a double-layer structure. The poor integrity of the specimens resulted in a weakened resistance to cracking. It is also notable that the splitting tensile strength of the specimens reduced as the thickness of the UML decreased. The low-temperature splitting tensile strength of the UMLs with 10 mm, 15 mm and 20 mm thickness was 2.463 MPa, 2.624 MPa and 2.662 MPa, respectively.

Figure 6 shows the failure strain of beam specimens with different UML thicknesses. As shown in Figure 6, it can be seen that the failure strain of the specimen decreased with decreasing UML thickness, which is in agreement with the results in Figure 5. This indicates that, the thinner the UML, the worse its low-temperature cracking resistance [26]. Despite having the worst low-temperature crack resistance, the failure strain of the 10 mm UML still met the requirement (≥2000 με) of the Technical Specifications for Construction of Highway Asphalt Pavements (JTG F40-2004) in China. Therefore, it was recommended to pave a 10 mm UML in order to save resources.

### 3.2. SiC Particle Size in UML

To investigate the effect of SiC particle size on the heating performance and deicing performance of the UML, the specimens were divided into four groups according to the particle sizes of the SiC partial replacement (20%) of limestone: 0# (control group), 1# (0~2.36 mm), 2# (0~9.5 mm), 3# (2.36~9.5 mm).

#### 3.2.1. Effect of SiC Particle Size on Temperature Rise Rate and Temperature Uniformity

The surface average temperature of each group of specimens after microwave heating was analyzed. The temperature rise rate of each group of specimens is shown in Figure 7. It can be seen from Figure 7 that the addition of SiC could effectively increase the temperature rise rate of the UML, with the larger particle size of SiC having a better increase effect on the temperature rise rate. The temperature rise rate of 3# specimens was 0.654 °C/s higher than that of 1# specimens. However, the large-particle-size (>2.36 mm) SiC had a very negative effect on the temperature uniformity of the UML. Figure 8 shows the range and standard deviation of the surface temperature after 30 s of microwave heating. As shown in Figure 8, the temperature range increased as the SiC particle size increased. It is also observed that the standard deviation of the specimens in groups 1#, 2# and 3# was 6, 16 and 25 times higher than that of the control specimens, respectively.

It can be seen from Figure 8 that, the larger the particle size of SiC, the worse the temperature uniformity of the UML. The following are the key reasons. The first is that SiC with a large particle size absorbed more microwave energy during microwave heating, resulting in a higher temperature than that for SiC with a small particle size [27]. Therefore, the UML with a larger particle size of SiC had a greater temperature range. Second, the SiC with the smaller particle size was more uniformly distributed and better dispersed in the UML. Figure 9 shows the distribution of different particle sizes of SiC in the UML, where the gray particles are SiC. It was discovered that the SiC with the large particle size in the UML was relatively dispersed and the contact between SiC particles was less, which is unfavorable to the rapid heat transfer between SiC particles. As a result, the UML with large-particle-size SiC had poor temperature uniformity. Third, the small-particle-size SiC had a larger specific surface area than the large-particle-size SiC, which allowed it to dissipate heat more quickly in the UML [28].

#### 3.2.2. Effect of SiC Particle Size on Deicing Time

According to the above results, an increase in SiC particle size increased the temperature rise rate while reducing the temperature uniformity of the UML. However, the temperature rise rate and temperature uniformity of the asphalt mixture can both affect the deicing efficiency. Therefore, to investigate the effect of SiC particle size on deicing time, a microwave deicing test was carried out at different initial temperatures. The deicing time of each group of specimens is shown in Figure 10.

As shown in Figure 10, it was found that the deicing time could be effectively shortened by adding SiC into the UML. However, the large-particle-size SiC in the UML increased the deicing time compared to the small-particle-size SiC in the UML. At an initial temperature of −10 °C, the deicing time for specimens in groups 1#, 2# and 3# was 47 s, 72 s and 82 s, respectively. This indicates that although the large-particle-size SiC in the UML could greatly increase the temperature rise rate, its poor temperature uniformity had a worse effect in reducing the deicing time than the small-particle-size SiC [29]. This was because when the temperature uniformity was poor, the ice in the higher-temperature areas melted while the ice in the lower-temperature areas was still frozen, which prevents the ice from separating from the road surface as a whole. The length of deicing time depended on the melting state of the ice in the low-temperature areas. Therefore, it is more beneficial to shorten the deicing time by adding small-particle-size SiC in the UML allowing ice to be removed from the road surface as a whole.

Additionally, the large-particle-size SiC could cause local overheating of the UML during microwave deicing. Table 4 shows the highest surface temperatures of specimens in groups 0#, 1#, 2# and 3# after microwave deicing. From Table 4, the highest surface temperatures of specimens in groups 2# and 3# could reach over 170 °C and 200 °C. The asphalt will soften under high temperatures, which causes damage to the pavement under the load of a microwave deicing vehicle [12,15].

In summary, the UML has an excessive temperature difference that can not only reduce the microwave deicing efficiency but also cause damage to the asphalt pavement. Therefore, SiC with particle sizes less than 2.36 mm was used to prepare the UML to improve the microwave deicing efficiency.

### 3.3. SiC Content in UML

The effect of SiC content on the surface temperature rise rate and deicing time of the UML was investigated under SiC particle size less than 2.36 mm. Three groups of UML specimens with SiC content of 10%, 20% and 30% of the total aggregate mass and control group specimens were prepared.

#### 3.3.1. Effect of SiC Content on Temperature Rise Rate

Figure 11 shows the temperature rise rate of each group of specimens obtained from the microwave heating test. It can be seen from Figure 11 that the temperature rise rate of the specimens increased as the SiC content increased. The temperature rise rate was 0.555 °C/s, 0.880 °C/s and 1.325 °C/s at 10%, 20% and 30% SiC content in the UML, respectively, which was 2.8 times, 4.4 times and 6.7 times higher than that of the control specimens. This is because in the same heating time, as the SiC content in the UML increased, more microwave energy was converted into heat, thus increasing the temperature rise rate of the UML. The UML increased the temperature rise rate of the specimens by nearly two times compared with the overall heating of the asphalt mixture added with SiC [21]. It can be concluded that the microwave heating efficiency of asphalt pavement can be effectively improved by accumulating the heat in the surface layer.

#### 3.3.2. Effect of SiC Content on Deicing Time

The microwave deicing test was carried out on each group of specimens. The deicing time for each group of specimens is shown in Figure 12. It can be seen that, as the initial temperature dropped, the deicing time grew longer. Additionally, it should be noted that the higher the SiC content, the lesser the deicing time at the same initial temperature. At an initial temperature of −10 °C, the deicing time for the specimens with 10%, 20% and 30% SiC was 54.7 s, 47 s and 44.7 s, respectively, which was 51%, 44% and 42% that of the control specimens. This indicates that the addition of SiC could effectively improve the microwave deicing efficiency of the UML. However, as the SiC content increased, the reduction in deicing time gradually decreased. For example, at an initial temperature of −10 °C, the deicing time for specimens with 30% SiC was only 2.3 s shorter than that with 20% SiC. Combined with the results in Section 3.3.1., it was discovered that the deicing time did not decrease proportionally as the temperature rise rate of the UML increased. This is because higher SiC content led to poorer temperature uniformity, which had a detrimental effect on reducing deicing time [29]. What is more, the opening holes on the surface of the UML specimens were filled with ice, the ice needed to melt off some mass before it could fall off the surface of the specimens under the influence of gravity, which limited the reduction of deicing time [27].

Figure 13 shows the mass of melted ice during microwave deicing of the UMLs with different SiC content. As can be seen in Figure 13, the mass of melted ice gradually decreased to a defined level as the SiC content increased. This indicates that, even though the temperature rise rate of the specimen was already high, the ice layer still needed to melt a certain mass to fall off the specimen. This is consistent with the above discussion. Furthermore, combined with Figure 11, it is clear that the higher the temperature rise rate of UML, the smaller the mass of melted ice. This is explained by the fact that, the higher the temperature rise rate of UML, the greater the temperature difference between the UML and the ice in the same heating time, and the greater the temperature difference, the smaller the adhesion strength between UML and ice, so the lesser the mass of ice needed to be melted and the shorter the deicing time.

In conclusion, since the increase of SiC content from 20% to 30% in the UML had little effect on improving the deicing efficiency, the content of SiC in the UML was determined as 20% from the perspective of resource saving and cost reduction.

### 3.4. Oil–Stone Ratio Design and Check

Based on the above results, the oil–stone ratio of the UML with 20% particle size less than 2.36 mm SiC was determined. The results of Marshall test and calculations are shown in Figure 14. It was observed that the porosity of the specimens decreased as the asphalt content increased. When the target void ratio was 6%, the corresponding oil–stone ratio was 7.4%. Therefore, the optimum oil–stone ratio of the UML was determined to be 7.4%. Table 5 shows the oil–stone ratio design inspection results. All properties of the UML met the requirements of the specification under the optimum oil–stone ratio of 7.4%, which means it had good road performance.

### 3.5. Energy Saving and Material Reduction Assessments

Compared with heating the whole asphalt pavement, the UML requires less energy and microwave-absorbing materials due to its higher heat utilization and thinness. Equations (8) and (9) were used to calculate the power and microwave-absorbing materials required for microwave deicing of the UML and traditional asphalt pavement on unit area pavement. The benefits of the UML in energy saving and material reduction were assessed through comparative analysis.
(8)$$P_{n}=\frac{c_{a}\rho hv}{1000}$$
(9)$$m_{SiC}=\rho h\times \frac{1}{1+s}\times a$$
where *P_n_* is the power required for microwave deicing per unit area (kW/m^2^), *h* is the thickness of the microwave-absorbing layer (m), *m_SiC_* is the mass of SiC added into the microwave-absorbing layer (kg), *s* is the oil–stone ratio of the microwave-absorbing layer and *a* is the ratio of SiC mass to total aggregate mass.

It is concluded from the above equations that, the thicker the microwave-absorbing layer, the more power and absorbing materials are required for microwave deicing. The values used in the calculations were obtained from the above results, as shown in Table 6. Calculations were performed for a 10 mm UML and a 40 mm conventional asphalt pavement upper layer. The power and amount of SiC required for microwave deicing per square meter of UML were calculated to be 22.55 kW and 4.34 kg, and the conventional asphalt pavement upper layer required 90.22 kW power and 17.36 kg SiC. In comparison, microwave deicing by an UML can save 75% of power and SiC. After the investigation and summary of the research results already reported, the power required for microwave deicing per square meter of different microwave-absorbing layers is shown in Table 7 [3,5,16]. As can be seen from Table 7, less power is required for microwave deicing by a UML. Therefore, the UML has a significant energy and material saving effect and has great advantages in both economic and environmental aspects.

## 4. Conclusions

A UML was designed to improve the microwave deicing efficiency for asphalt pavement. The thickness, SiC particle size, SiC content and oil–stone ratio of the UML were determined. In addition, the energy saving and material reduction effect of the UML was evaluated. The main conclusions obtained are as follows:The thickness of the UML was influenced by ambient temperature and microwave absorption performance of UML, which had been calculated to be 10~23 mm. The thickness of the UML was recommended to be 10 mm under the premise of ensuring economic performance and crack resistance.The influence of SiC particle size on the temperature rise rate and the temperature uniformity of the UML was different. The larger the SiC particle size, the higher the temperature rise rate. A UML with a SiC particle size greater than 2.36 mm had a 0.654 °C/s higher temperature rise rate than a UML with a SiC particle size less than 2.36 mm. However, as the SiC particle size increased, the temperature uniformity of the UML worsened. The temperature standard deviation of UML with SiC particle size greater than 2.36 mm was four times greater than that of UML with SiC particle size less than 2.36 mm. Moreover, the poor temperature uniformity was not conducive to a shorter deicing time. Therefore, using SiC with a particle size less than 2.36 mm to prepare a UML can reduce the negative effect of temperature inhomogeneities on deicing efficiency and avoid local overheating damage to asphalt pavements.The increase in SiC content had a positive effect on improving the temperature rise rate and shortening the deicing time of the UML. The temperature rise rate of the UML with 20% SiC was increased by 3.4 times compared with that of control group, and the deicing time was shortened by 56%. After taking into account the deicing efficiency and material cost, the ideal SiC content in the UML was determined to be 20%.The optimum oil–stone ratio for the UML was 7.4% at a target void ratio of 6%, and the UML had good road performance.Compared with heating the whole asphalt pavement, heating the UML can save 75% microwave energy and microwave-absorbing material and has good economic benefits.

## Figures and Tables

**Figure 1 materials-16-03080-f001:**
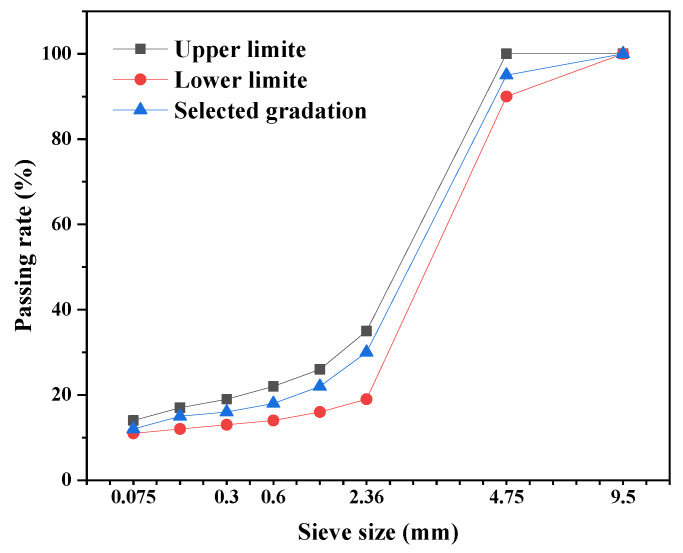
Gradation curve of asphalt mixture for the UML.

**Figure 2 materials-16-03080-f002:**
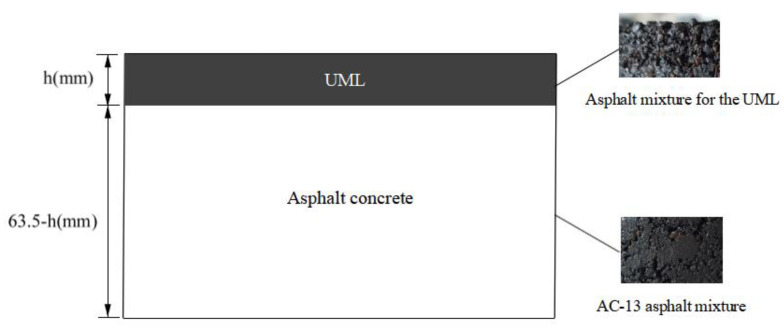
Structure of the low-temperature splitting test specimens.

**Figure 3 materials-16-03080-f003:**
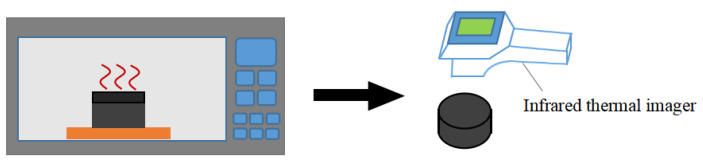
The process of microwave heating test.

**Figure 4 materials-16-03080-f004:**
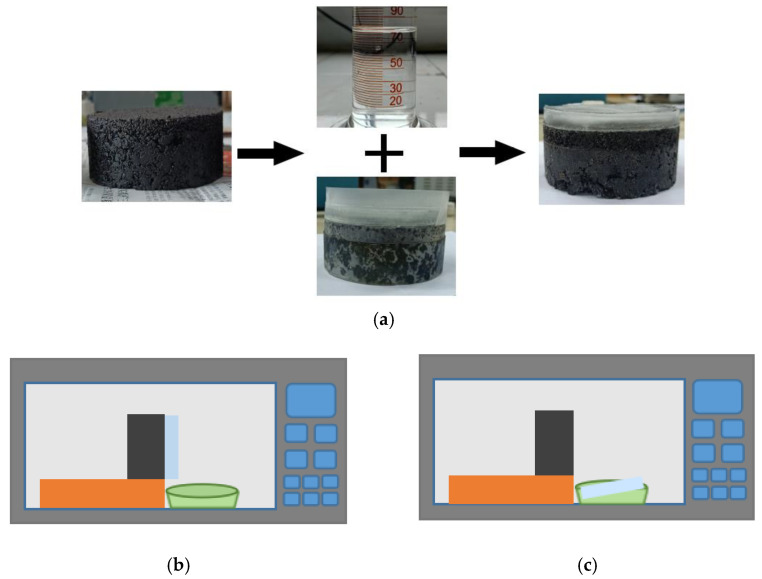
The process of microwave deicing test: (**a**) ice layer preparation; (**b**) ice layer melting; (**c**) ice layer falling off.

**Figure 5 materials-16-03080-f005:**
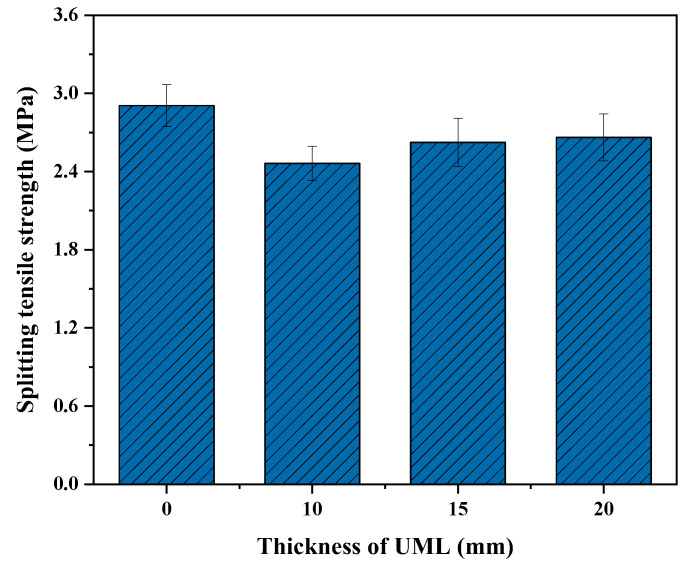
Splitting tensile strength of UMLs with different thicknesses.

**Figure 6 materials-16-03080-f006:**
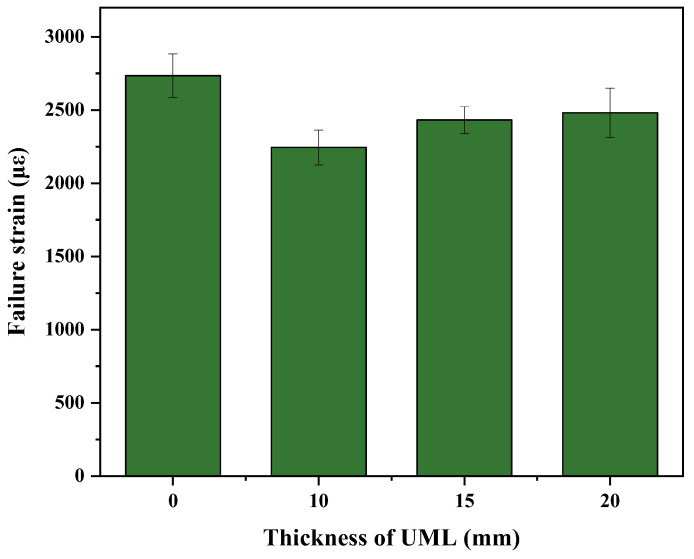
Failure strain of UMLs with different thicknesses.

**Figure 7 materials-16-03080-f007:**
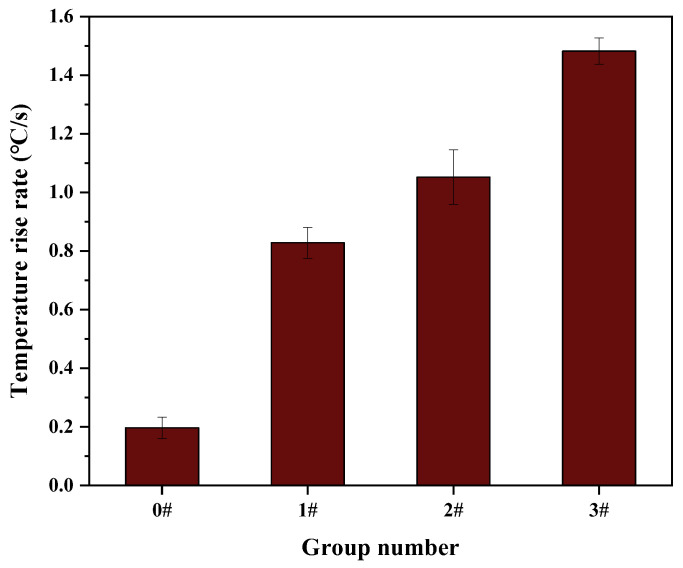
Temperature rise rate of UMLs with different SiC particle sizes.

**Figure 8 materials-16-03080-f008:**
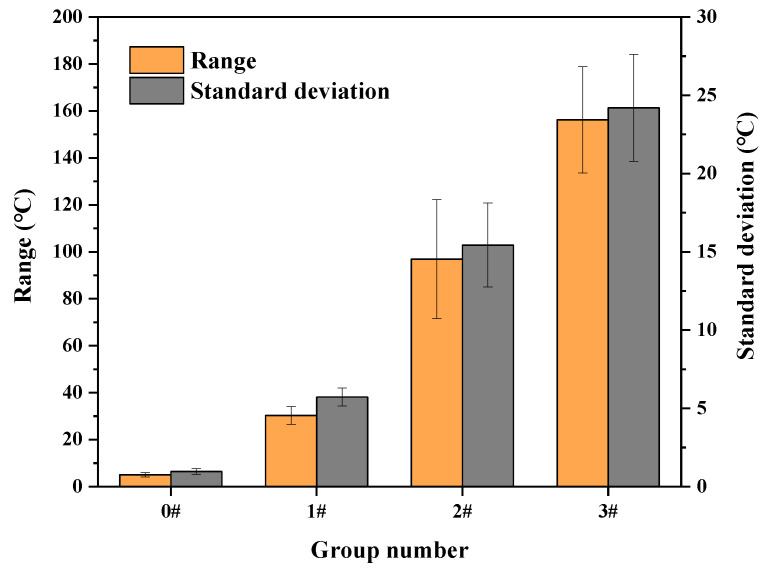
Temperature range and standard deviation of UMLs with different SiC particle sizes.

**Figure 9 materials-16-03080-f009:**
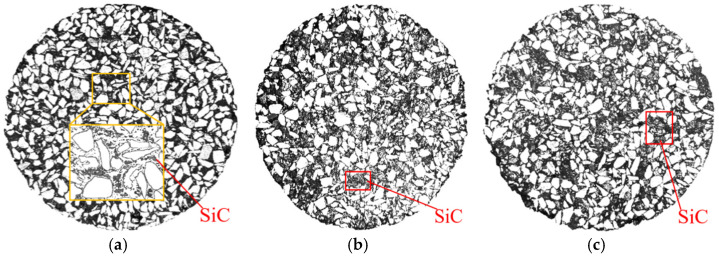
Distribution of SiC in (**a**) 1#; (**b**) 2#; (**c**) 3# specimens.

**Figure 10 materials-16-03080-f010:**
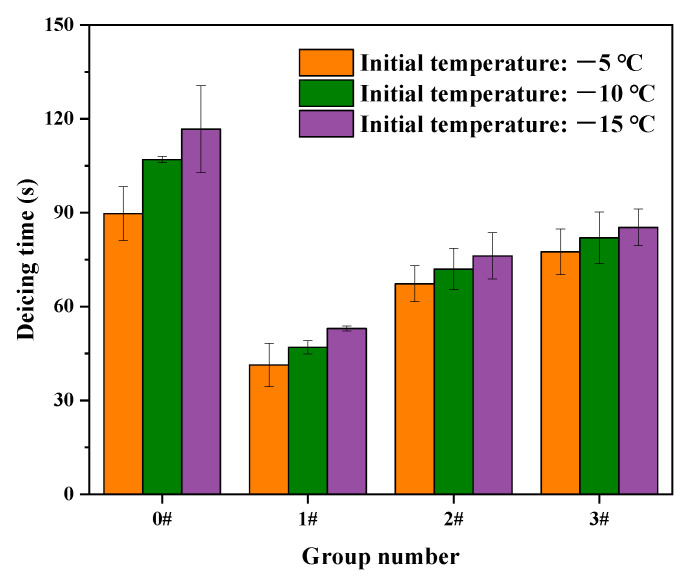
Deicing time of UMLs with different SiC particle sizes.

**Figure 11 materials-16-03080-f011:**
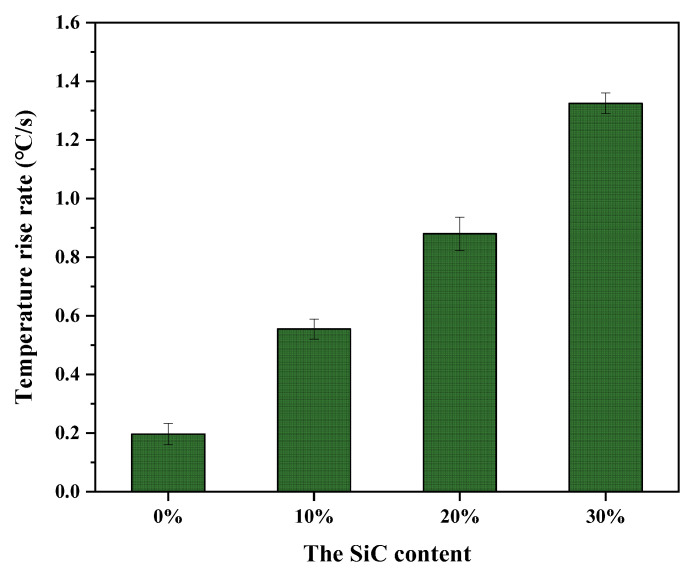
Temperature rise rate of UMLs with different SiC contents.

**Figure 12 materials-16-03080-f012:**
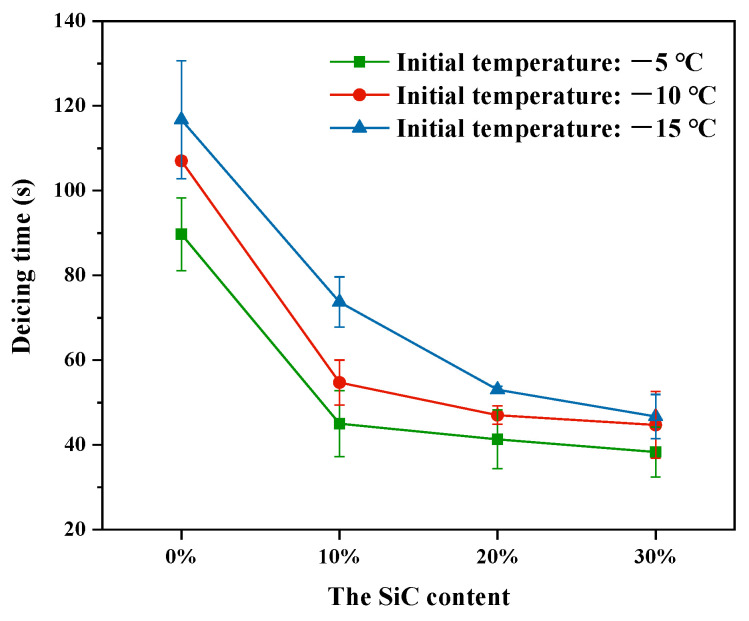
Deicing time of UMLs with different SiC contents.

**Figure 13 materials-16-03080-f013:**
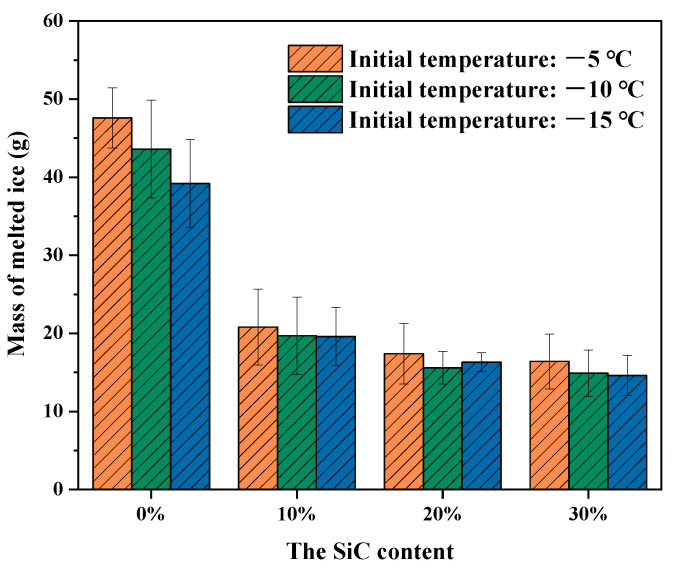
Mass of melted ice of UMLs with different SiC contents.

**Figure 14 materials-16-03080-f014:**
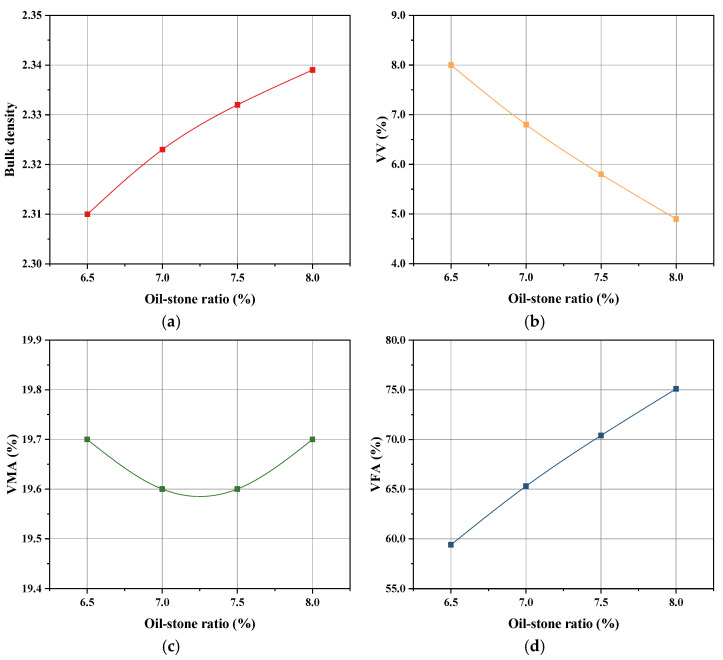
Blk density (**a**); VV (**b**); VMA (**c**); VFA (**d**) under different oil–stone ratios.

**Table 1 materials-16-03080-t001:** Properties of bitumen.

Properties	Unit	Test Results	Specification
25 °C penetration	0.1 mm	44	40–70
5 °C ductility	cm	32	≥30
Softening point	°C	91.0	≥90
165 °C dynamic viscosity	Pa·s	2.84	≤3
Elastic recovery	%	96	≥95

**Table 2 materials-16-03080-t002:** Properties of limestone.

Properties	Unit	Test Results	Specification
Crushing value	%	22.7	≤26.0
Abrasion value	%	19.5	≤28.0
Apparent specific gravity	g/cm^3^	2.71	≥2.60
Water absorption	%	0.12	≤2.00
Adhesion with asphalt	-	5	≥5
Polished stone value	BPN	42	≥42

**Table 3 materials-16-03080-t003:** Values in Equations (3)–(5).

*c_a_*/J/(kg·°C)	*v*/°C/s	*m_i_*/kg/m^2^	*ρ*/kg/m^3^
1100	0.98	1.834	2.4 × 10^3^

**Table 4 materials-16-03080-t004:** Highest temperature of UMLs with different SiC particle sizes.

Initial Temperature/°C	0#/°C	1#/°C	2#/°C	3#/°C
−5	26.3	41.9	172.5	204.1
−10	27.8	52.7	180.7	209.3
−15	28.5	60.3	186.5	211.3

**Table 5 materials-16-03080-t005:** Oil–stone ratio design inspection results.

Properties	Unit	Test Results	Specification
Drainage loss	%	0.03	≤0.1
Cantabro loss	%	5.4	≤15
Residual stability	%	94.1	≥85
Freeze–thaw splitting strength ratio	%	91.1	≥80
High-temperature rutting deformation	mm	0.09	≤3

**Table 6 materials-16-03080-t006:** Values in Equations (8) and (9).

*ρ*/kg/m^3^	*v*/°C/s	*s*	*a*
2.33 × 10^3^	0.880	7.4%	20%

**Table 7 materials-16-03080-t007:** Power required for different microwave-absorbing layers.

Microwave-Absorbing Layer	Unit	Power
UML	kW	22.55
SiC-Fe_3_O_4_ microwave-enhanced functional layer	kW	87.94
Acrylic superhydrophobic asphalt pavement coating	kW	41.94
Asphalt mixture with activated carbon powder filler	kW	72.43

## Data Availability

All data generated or analyzed during this study are included in this published article.

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
