# Peer review of "An Ultra-Thin, Microwave-Absorbing Wear Layer for Pavement Deicing"

_materials, 2023, doi:10.3390/ma16083080_

Round 1

Reviewer 1 Report

Figure 1 is not required as it is not adding any information except particle size. The particle size may be given in the text.

The thickness of the ice layer is not given in the manuscript. 

The thickness range of UML will be a function of SiC also. In case the 4.5-9.5 mm SiC will be used, the UML thickness will be a minimum of 9.5 mm. This should be explained.

Results & discussion are highly confusing. It is not clear which parameter was optimized first and why?

Author Response

Response to Reviewer 1 Comments

Journal: Materials

Manuscript ID: materials-2297836

Title: An Ultra-Thin Microwave Absorbing Wear Layer for Pavement Deicing

Dear reviewer:

Thank you very much for your decision and constructive comments on our manuscript. Your suggestions are really helpful for improving our manuscript. We have carefully considered the suggestions and made some changes. All changes have been marked using the "track mark" function. According to your suggestions, we have made the following revisions to this manuscript:

Point 1: Figure 1 is not required as it is not adding any information except particle size. The particle size may be given in the text.

Response 1: Thank you very much for your advice. We respect and understand your views, however, Figure 1 not only shows the particle size of SiC but also shows the appearance of SiC. It is because SiC has a very similar appearance to aggregates that we believe it is feasible to use SiC as a substitute for aggregates in asphalt mixtures to prepare UML.

Point 2: The thickness of the ice layer is not given in the manuscript.

Response 2: Thank you very much for your advice. We are very sorry that we did not obviously give the thickness of the ice layer in the manuscript, as it is not the focus of this work. However, we gave the thickness of the ice layer on page 5, line 156. The thickness of the ice layer is 1 cm.

Point 3: The thickness range of UML will be a function of SiC also. In case the 4.5-9.5 mm SiC will be used, the UML thickness will be a minimum of 9.5 mm. This should be explained.

Response 3: Thank you very much for your advice. Following your suggestion, we have added a note on this issue on page 7, lines 232-234. The sentence reads as follows: ”Based on the layer thickness range, UML used the gradation shown in Figure 2 with a maximum aggregate particle size of 9.5 mm.” In this work, we first calculated the layer thickness range of UML. Based on this range, the UML gradation was selected. Therefore, in the later study, the maximum particle size of SiC in the UML was 9.5 mm.

Point 4: Results & discussion are highly confusing. It is not clear which parameter was optimized first and why?

Response 4: Thank you very much for your advice. We are sorry for not clarifying the UML prepare process in our manuscript. The text order in the Results and Discussions section is the order of optimization of the UML parameters. The following is the order of optimization of the UML parameters: 1. layer thickness, 2. SiC particle size, 3. SiC content, 4. oil-stone ratio. The reason we determined the layer thickness of the UML first is that the UML gradation is related to the layer thickness. Once the gradation of UML has been determined, the particle size range of the aggregates in UML is determined. Compared to the content of SiC in UML, we determined the particle size of SiC in UML first. This is because, in the course of the tests, we found that the particle size of SiC has a significant influence on the temperature uniformity of UML. Only by first determining the SiC particle size in UML can the adverse impact of temperature inhomogeneities of the UML on deicing time be reduced. After determining the content of SiC in UML, the oil-stone ratio of UML was finally determined.

Thank you again for your suggestions. If there are any other modifications we could make, we would like very much to modify them and we really appreciate your help.

Reviewer 2 Report

In this paper, the microwave heating mechanism of SiC fille Bitumen materials were represented. Overall this paper can be considered for publication but there are some points that need to be improved. 

1. Abstract: Please efine materials clearly. I mean if filler is SiC explain what matrix is. for example SiC filled .... based materials.

2. Introduction: Which materials properties effect to the microwave heating mechanism. For example dielectric and magnetic properties? This mechanism works as an electromagnetic wave absorption mechanism. I suggest you to add some information to it. Bellow reference can help you.

AKINAY, Y., GUNES, U., Çolak, B., & Cetin, T. (2022). Recent progress of electromagnetic wave absorbers: A systematic review and bibliometric approach. ChemPhysMater.

Akinay, Y. (2019). Microwave-Absorbing Properties of Single-and Multilayer Materials: Microwave-Heating Mechanism and Theory of Material–Microwave Interaction. In Nanoscale Networking and Communications Handbook (pp. 411-424). CRC Press.

3. Materials Method: The preparation of SiC-filled bitumen is not provided. How did you prepare these samples? What was the filler rate? How did you mix them?

Author Response

Response to Reviewer 2 Comments

Journal: Materials

Manuscript ID: materials-2297836

Title: An Ultra-Thin Microwave Absorbing Wear Layer for Pavement Deicing

Dear reviewer:

Thank you very much for your decision and constructive comments on our manuscript. Your suggestions are really helpful for improving our manuscript. We have carefully considered the suggestions and made some changes. All changes have been marked using the "track mark" function. According to your suggestions, we have made the following revisions to this manuscript:

Point 1: Abstract: Please efine materials clearly. I mean if filler is SiC explain what matrix is. for example SiC filled .... based materials.

Response 1: Thank you very much for your advice. Following your suggestion, we have added some information about this on page 1, lines 9-11. The sentence reads as follows: “To improve the utilization efficiency of microwave energy and the deicing efficiency, using silicon carbide (SiC) replaced aggregates in asphalt mixtures to prepare an ultra-thin microwave absorbing wear layer (UML).”

Point 2: Introduction: Which materials properties effect to the microwave heating mechanism. For example dielectric and magnetic properties? This mechanism works as an electromagnetic wave absorption mechanism. I suggest you to add some information to it. Bellow reference can help you.

AKINAY, Y., GUNES, U., Çolak, B., & Cetin, T. (2022). Recent progress of electromagnetic wave absorbers: A systematic review and bibliometric approach. ChemPhysMater.

Akinay, Y. (2019). Microwave-Absorbing Properties of Single-and Multilayer Materials: Microwave-Heating Mechanism and Theory of Material–Microwave Interaction. In Nanoscale Networking and Communications Handbook (pp. 411-424). CRC Press.

Response 2: Thank you very much for your advice. We have checked the contents of the Introduction carefully and added more information on microwave absorbing materials into the Introduction part of the revised manuscript on page 2, lines 59-61. And we have added references [14] and [15] to support this idea.

  1. Akinay, Y.; Gunes, U.; Çolak, B.; Cetin, T. Recent Progress of Electromagnetic Wave Absorbers: A Systematic Review and Bibliometric Approach. ChemPhysMater. 2022.
  2. Akinay, Y. Microwave-Absorbing Properties of Single-and Multilayer Materials: Microwave-Heating Mechanism and Theory of Material–Microwave Interaction. In Nanoscale Networking and Communications Handbook; CRC Press, 2019; pp. 411-424.

Point 3: Materials Method: The preparation of SiC-filled bitumen is not provided. How did you prepare these samples? What was the filler rate? How did you mix them?

Response 3: Thank you very much for your advice. Following your suggestion, we have added some information about this on page 5, lines 140-141. The sentence reads as follows: “The UML was prepared by mixing SiC in different particle sizes and contents with bitumen, aggregate, and mineral powder.” We feel sorry that we did not provide a detailed description of the specimen preparation process in the manuscript. This is because the process of preparing UML was the same as that of preparing a normal asphalt mixture. The only difference is that part of the aggregates in UML was replaced with SiC. The SiC content in the specimens was 0%, 10%, 20%, and 30%, respectively. After investigating the impact of SiC content on deicing time, the SiC content in UML was determined to be 20%.

Thank you again for your suggestions. If there are any other modifications we could make, we would like very much to modify them and we really appreciate your help.

Reviewer 3 Report

The manuscript entitled “An Ultra-Thin Microwave Absorbing Wear Layer for Pavement Deicing”. It's an interesting topic in academic research that works with microwave radiation. That said, the manuscript is well written with results well discussed in an understandable way. Therefore, I recommend acceptance of the same in Journal Materials. However, I have a few small suggestions and comments before final acceptance:

Moreover, accumulating heat on the surface also has the effect of saving energy and reducing emissions. Is it suggestible to specify which emissions would be those in this frese?

I suggest you name what the acronym SiC means the first time it is mentioned in the manuscript.

When it is stated that the UML has a significant energy and material saving effect and has great advantages in economic and environmental aspects. It is strongly suggested that they place a comparative table with results already reported in the literature, so that the reader can follow and compare these significant UML results.

Compared with heating the whole asphalt pavement, heating the UML can save 75% microwave energy and microwave absorbing material and has good economic benefits. Suggest that you put in a table the economic benefits compared to those reported in others already reported. Which makes it more interesting for readers.

Why did the study not evaluate other output powers higher or lower than 900W? Because microwaves demand energy consumption.

Author Response

Response to Reviewer 3 Comments

Journal: Materials

Manuscript ID: materials-2297836

Title: An Ultra-Thin Microwave Absorbing Wear Layer for Pavement Deicing

Dear reviewer:

Thank you very much for your decision and constructive comments on our manuscript. Your suggestions are really helpful for improving our manuscript. We have carefully considered the suggestions and made some changes. All changes have been marked using the "track mark" function. According to your suggestions, we have made the following revisions to this manuscript:

Point 1: Moreover, accumulating heat on the surface also has the effect of saving energy and reducing emissions. Is it suggestible to specify which emissions would be those in this frese?

Response 1: Thank you very much for your advice. Following your suggestion, we have added some information about this on page 2, lines 51-52. The sentence reads as follows: ”Moreover, accumulating heat on the surface also has the effect of saving energy and reducing emissions. Liu accumulated the heat on the pavement surface by adding waste steel wool into the asphalt mixture which saved the energy required for microwave deicing and reduced the emissions of SO2 and NOX.”

Point 2: I suggest you name what the acronym SiC means the first time it is mentioned in the manuscript.

Response 2: Thank you very much for your advice. We have explained the meaning of SiC when it was first mentioned in the manuscript. The sentence reads as follows: “To improve the utilization efficiency of microwave energy and the deicing efficiency, using silicon carbide (SiC) replaced aggregates in asphalt mixtures to prepare an ultra-thin microwave absorbing wear layer (UML).”

Point 3: When it is stated that the UML has a significant energy and material saving effect and has great advantages in economic and environmental aspects. It is strongly suggested that they place a comparative table with results already reported in the literature, so that the reader can follow and compare these significant UML results.

Response 3: Thank you very much for your advice. We think this is an excellent suggestion. Following your suggestion, we have added a comparative table (Table 7) of the power required for microwave deicing of different microwave absorbing layers on page 15, line 433. The data in Table 7 were calculated from the reported results. Table 7 is shown below.

Table 7. Power required for different microwave absorbing layers.

Microwave absorbing layer

Unit

Power

UML

kW

22.55

SiC-Fe3O4 Microwave Enhanced Functional Layer

kW

87.94

Acrylic Superhydrophobic Asphalt Pavement Coating

kW

41.94

Asphalt Mixture with Activated Carbon Powder Filler

kW

72.43

Although we have added a table comparing the power required for UML and other microwave absorbing layers, we regret that we have not added a table comparing the microwave absorbing materials required for UML and other microwave absorbing layers. This is because it is difficult to obtain the relevant parameters of the microwave absorbing material in the reported results, so we cannot accurately calculate the mass of the microwave absorbing material in other microwave absorbing layers. For example, in Liu's study [17], we can obtain the volume fraction of activated carbon powder in the filler, but we cannot obtain the density of activated carbon powder and the mass fraction of filler in the asphalt mixture. Therefore, we cannot accurately calculate the mass fraction of activated carbon powder in the asphalt mixture.

  1. Liu, Z.; Yang, X.; Wang, Y.; Luo, S. Engineering Properties and Microwave Heating Induced Ice-Melting Performance of Asphalt Mixture with Activated Carbon Powder Filler. Build. Mater.2019, 197, 50-62.

Point 4: Compared with heating the whole asphalt pavement, heating the UML can save 75% microwave energy and microwave absorbing material and has good economic benefits. Suggest that you put in a table the economic benefits compared to those reported in others already reported. Which makes it more interesting for readers.

Response 4: Thank you very much for your advice. We think this is an excellent suggestion and really helpful for improving our manuscript. However, because it is difficult to obtain the density and cost of different microwave absorbing materials from the reported results, it is not possible to accurately calculate the cost required for other microwave absorbing layers. It is therefore difficult to accurately compare the economics of UML with other reported microwave absorbing layers.

Point 5: Why did the study not evaluate other output powers higher or lower than 900W? Because microwaves demand energy consumption.

Response 5: Thank you very much for your question. Your question is quite reasonable. The output power of the microwave oven used in this work was a fixed 900 W. Due to the limitations of the test equipment, only the output power of 900 W has been studied in this work. Your suggestion is very valuable and we will investigate the effect of power on the deicing performance of UML in the next work.

Thank you again for your suggestions. If there are any other modifications we could make, we would like very much to modify them and we really appreciate your help.

Round 2

Reviewer 1 Report

Figure 1 is not required as it has no significance. Mention the particle size in the text.

Kindly mention the units of dimensions in Figure 3.

Kindly follow the SI units. In general, the distance is reported in mm or m, not in cm. 

Line 126: Which equipment was used and what specifications?

Line 201 Replace dates with 'data'.

Line 245: What are engineering application requirements? Please cite and report the minimum crack resistance recommended.

Line 257: how can you write significant change without any statistical test?

Error bars in Figure 8 are presenting what? What do you mean by temperature range?

English editing is required particularly in the R&D section. 

Define deicing time clearly. It is not clear whether the deicing time includes the microwave treatment time or not.

Section 3.4: At what location the temperature of the specimen was measured? Microwave heating is never uniform in the system used for conducting the tests. Keeping this point in view, rewrite Section 3.4.

Limit the references to 30.

Author Response

Response to Reviewer 1 Comments

Journal: Materials

Manuscript ID: materials-2297836

Title: An Ultra-Thin Microwave Absorbing Wear Layer for Pavement Deicing

Dear reviewer:

Thank you very much for your decision and constructive comments on our manuscript. Your suggestions are really helpful for improving our manuscript. We have carefully considered the suggestions and made some changes. All changes have been marked using the "track mark" function. According to your suggestions, we have made the following revisions to this manuscript:

Point 1: Figure 1 is not required as it has no significance. Mention the particle size in the text.

Response 1: Thank you very much for your advice. We have removed Figure 1 from the manuscript and added information on SiC particle size to the text. The sentence reads as follows: “After sieving, three different particle sizes of SiC were obtained: 0~2.36 mm, 0~9.5 mm and 2.36~9.5 mm.” (Page 3, lines 96-97).

Point 2: Kindly mention the units of dimensions in Figure 3.

Response 2: Thank you very much for your advice. Following your suggestion, we have added dimension units to Figure 3 (modified to Figure 2).

Point 3: Kindly follow the SI units. In general, the distance is reported in mm or m, not in cm.

Response 3: Thank you very much for your advice. We feel sorry for our carelessness. We have changed the units of distance in the text to the SI units.

Point 4: Line 126: Which equipment was used and what specifications?

Response 4: Thank you very much for your advice. We have indicated the equipment used and the specification in section 2.2.2 of the manuscript.

Point 5: Line 201 Replace dates with 'data'.

Response 5: Thank you very much for your advice. We sincerely thank the reviewer for careful reading. As suggested by the reviewer, we have corrected the "dates" into "data".

Point 6: Line 245: What are engineering application requirements? Please cite and report the minimum crack resistance recommended.

Response 6: Thank you very much for your advice. We apologize for the misrepresentation here. The engineering application requirements here refer to the minimum requirements for the splitting tensile strength of asphalt mixtures. The minimum requirement of 2.00 MPa is a citation from a master's thesis, but it is not cited in the manuscript because it is not yet publicly available. To fully verify the low temperature crack resistance of UML, we have added test results from the low temperature bending test to the manuscript (Figure 6). The results of the low-temperature bending test indicate that the failure strain of 10 mm UML met the specification requirement (≥2000με). (Page 8, lines 268-274).

Point 7: Line 257: how can you write significant change without any statistical test?

Response 7: Thank you very much for your advice. We apologize for using the wrong word here. We have changed the expression of this sentence. The sentence reads as follows: “It can be seen from Figure 7 that the addition of SiC could effectively increase the temperature rise rate of the UML, with the larger particle size of SiC having a better increase effect on the temperature rise rate.” (Page 9, lines 287-288).

Point 8: Error bars in Figure 8 are presenting what? What do you mean by temperature range?

Response 8: Each data in the manuscript was the average of the test results of three specimens. The error bars indicate the standard deviation of three specimen results. The large standard deviation of the 2# test results in Figure 8 was due to the fact that the SiC content in UML is only 20%, and the particle size range of SiC was relatively large (0~9.5mm), while the size of the specimen was limited. As a result, the distribution and content of SiC in each specimen were not exactly the same. Furthermore, the larger the particle size of SiC, the greater its microwave absorption ability, leading to large differences in the test results of different specimens. The temperature range is the difference between the highest and lowest temperatures on the surface of the specimen.

Point 9: English editing is required particularly in the R&D section.

Response 9: Thank you very much for your advice. About the language, we have discussed this with a native speaker in our group and a lot of revisions have been made to this paper. And if the language is still worse than your requirement, we prefer to use the Language Editing Services when the content of our paper can meet the requirement of publishing, to make this paper a better publication. We believe that the Language Editing Services could improve our paper a lot in the language field.

Point 10: Define deicing time clearly. It is not clear whether the deicing time includes the microwave treatment time or not.

Response 10: Thank you very much for your advice. Following your suggestion, we have clearly defined the deicing time in the manuscript. (Page 6, lines 180-185). The sentences read as follows: “The microwave oven was turned off when the ice layer fall off the specimens. The time taken for the microwave oven to heat the specimens was the time required for the ice layer to fall off. The time taken for the ice to fall off, the mass of melted ice and the temperature of the specimen surface was recorded. Consider the time taken for the ice to fall off as deicing time.”

Point 11: Section 3.4: At what location the temperature of the specimen was measured? Microwave heating is never uniform in the system used for conducting the tests. Keeping this point in view, rewrite Section 3.4.

Response 11: Thank you very much for your advice. We measured the temperature on the upper surface of the specimen. Following your suggestion, we have deleted section 3.4 and added the content from section 3.4 to sections 3.2.2 and 3.3.2 after rewriting it.

Point 12: Limit the references to 30.

Response 12: Thank you very much for your advice. Following your suggestion, we have limited the number of references in the manuscript to 30.

Thank you again for your suggestions. If there are any other modifications we could make, we would like very much to modify them and we really appreciate your help.